# Adherence and factors influencing adherence to glaucoma medications among adult glaucoma patients in Ethiopia: A systematic review and meta-analysis

Kibruyisfaw Weldeab Abore [1,2]*, Estifanos Bekele Fole[3], Mahlet Tesfaye Abebe[3], Natnael Fikadu Tekle[4], Robel Bayou Tilahun[4], Fraol Daba Chinkey[5], Michael Teklehaimanot Abera[6]

1 Department of Ophthalmology, St. Paul's Hospital Millennium Medical College, Addis Ababa, Ethiopia, 2 Department of Pediatrics, Yirgalem Hospital Medical College, Yirgalem, Sidama, Ethiopia, 3 Department of Medicine, Yirgalem Hospital Medical College, Yirgalem, Sidama, Ethiopia, 4 Long Term Care Department, Burjeel Medical City, Abu Dhabi, United Arab Emirates, 5 Bin Haider Healthcare Center, Dubai, United Arab Emirates, 6 Department of Radiology, College of Health Sciences, Addis Ababa University, Addis Ababa, Ethiopia

* Kibruyisfaww@gmail.com

**Data Availability Statement:** All relevant data are within the manuscript and its Supporting information files.

## Abstract

### Background

Intraocular pressure is the only modifiable risk factor for the development and progression of glaucoma. Raised intraocular pressure could cause progressive visual field loss and blindness if left uncontrolled. Adherence to ocular hypotensive medications is vital to prevent optic nerve damage and its consequences. This study was conducted to systematically summarize the magnitude of glaucoma medication adherence and factors influencing adherence to glaucoma medications among adult glaucoma patients in Ethiopia.

### Methods

Database searches to identify research articles were conducted on PubMed, EMBASE, Cochrane, AJOL, SCOPUS, and Google Scholar without restriction on the date of publication. Data extraction was done using a data extraction Excel sheet. Analysis was performed using STATA version 16. Heterogeneity was assessed using $I^2$ statistics. Pooled prevalence and pooled odds ratio with a 95% confidence interval using a random effect model were computed.

### Result

We included six studies with a total of 2101 participants for meta-analysis. The magnitude of adherence to glaucoma medication was found to be 49.46% (95% CI [41.27–57.66]). Urban residents (OR = 1.89, 95% CI; 1.29–2.49) and those with normal visual acuity (OR = 2.82, 95% CI; 0.85–4.80) had higher odds of adherence to glaucoma medications. Patients who

**Funding:** The author(s) received no specific funding for this work.

**Competing interests:** The authors have declared that no competing interests exist.

pay for the medications themselves (OR = 0.22, 95% CI; 0.09–0.34) were found to have 78% lower odds of adherence than their counterparts.

## Conclusion

The magnitude of glaucoma medication adherence is lower than expected. Place of residence, visual acuity, and payment means had statistically significant associations with glaucoma medication adherence. Tailored health education on medication adherence and subsidization of glaucoma medication is recommended.

## Introduction

Glaucoma is a progressive optic neuropathy, and it is one of the leading causes of preventable blindness [1]. Globally, it is estimated to affect around 76 million individuals and cause blindness in 7.7 million individuals [2]. Previous studies have shown that there are various risk factors to develop glaucoma. High intraocular pressure (IOP) is a modifiable risk factor that is a target of treatment among glaucoma patients [3]. Currently available treatment modalities that include medical, surgical, and laser therapy are all primarily designed to lower the IOP [4].

Successful medical treatment of patients with glaucoma is contingent upon the adherence of the patients to the prescribed medication [5, 6]. Adherence is the degree to which the patient takes the prescribed medication according to the prescription given by a health professional [5]. Adherence to glaucoma medication has been shown to decrease the progressive visual field loss and subsequent loss of vision among glaucoma patients [7, 8].

Although it is highly unreliable and prone to bias, self-reporting is the most commonly used method of assessing adherence [9, 10]. Previous studies done in different parts of the world have reported a highly variable level of non-adherence ranging from 5% to 80% [11]. Furthermore, studies done in Ethiopia have also reported levels of adherence ranging from 32.5% [12] to 61.4% [13]. Some of the factors associated with adherence to glaucoma medication include sociodemographic factors, poor understanding of the disease and its course, baseline clinical condition, and medication related factors such as dosing, side effects, and costs of medication [14–18].

In Ethiopia, few studies have assessed the magnitude of adherence and the factors influencing adherence to glaucoma medications among adult glaucoma patients. Furthermore, the findings reported in various studies conducted in Ethiopia vary greatly. This necessitates pooling of available research findings to make evidence based recommendations. Therefore, this study aims to synthesize the pooled overall magnitude of glaucoma medication adherence and factors influencing adherence to glaucoma medication among adult glaucoma patients in Ethiopia.

## Methods and materials

### Study design and study setting

A systematic review and meta-analysis were done to assess the magnitude of glaucoma medication adherence and factors influencing adherence to glaucoma medications among adult glaucoma patients in Ethiopia. The protocol for the systematic review was registered on PROSPERO (Registration number: CRD42023449004).

## Search strategy

A comprehensive and systematic search of articles from PubMed, EMBASE, Cochrane, AJOL, SCOPUS, and Google Scholar databases was made using the search terms (("anti-glaucoma" OR "glaucoma medication" OR "glaucoma drug" OR "topical anti-glaucoma") AND (adherence OR compliance)) AND ("Ethiopia"). The full PubMed search was ("anti-glaucoma"[All Fields] OR "glaucoma medication"[All Fields] OR "glaucoma drug"[All Fields] OR "topical anti-glaucoma"[All Fields]) AND ("adharance"[All Fields] OR "adhere"[All Fields] OR "adhered"[All Fields] OR "adherence"[All Fields] OR "adherences"[All Fields] OR "adherent"[All Fields] OR "adherents"[All Fields] OR "adherer"[All Fields] OR "adherers"[All Fields] OR "adheres"[All Fields] OR "adhering"[All Fields] OR ("compliances"[All Fields] OR "patient compliance"[MeSH Terms] OR ("patient"[All Fields] AND "compliance"[All Fields]) OR "patient compliance"[All Fields] OR "compliance"[All Fields] OR "compliance"[MeSH Terms])) AND "Ethiopia"[All Fields]. The search was made up to September 2023, with restriction to only articles published in English. The Preferred Reporting Items for Systematic Reviews and Meta-Analyses (PRISMA) guideline [19] was used to document and report the steps of study selection.

## Eligibility criteria

**Inclusion criteria.** This review included studies (cross-sectional, cohort, and case-control studies) that evaluated glaucoma medication adherence and/or factors influencing adherence to glaucoma medications among adult glaucoma patients in Ethiopia with no restriction on the year of publication of the study.

**Exclusion criteria.** Studies conducted outside of Ethiopia and studies that were not published in English were excluded from the study.

## Quality appraisal

Quality assessment of the included studies was done using the Newcastle-Ottawa scale (NOS) by three reviewers independently [20]. It assesses bias in three domains, namely selection, comparability, and outcome domains. Those with a grade of less than 7 were assessed as poor, and those with greater than or equal to 7 were assessed as having good quality. Disagreements in scoring between reviewers were settled through discussion with the help of a fourth reviewer.

## Data extraction

Two reviewers searched databases and applied the eligibility criteria independently to select studies. Covidence was used to manage articles. Examination of the selected studies against the eligibility criteria was done by two reviewers. Data extraction from selected studies was done independently by two reviewers, and examination of the extracted data was done by a third reviewer. Differences between the reviewers were resolved through discussion, and when agreement was not reached, a third reviewer was involved to mediate. A pre-prepared EXCEL data extraction sheet was utilized to extract data from included studies, which included the name of the primary author, region of the country, year of publication, study design, sample size, magnitude of adherence to anti-glaucoma medication, and risk factors with the corresponding measure of effect.

## Data analysis

Data was exported from Excel to STATA v.16 for analysis. The pooled magnitude of adherence and pooled odds ratio with a 95% confidence interval were computed using random effect model with the DerSimonian-Laird method and presented using a forest plot. To calculate the pooled odds ratio, we included variables that were reported as a statistically significant variable in at least two studies. Heterogeneity was assessed using $I^2$ statistics, and meta-regression was done to identify potential sources of heterogeneity. Furthermore, subgroup analysis was conducted based on the method of adherence ascertainment used by the studies. Publication bias was assessed using a funnel plot and Eggers test.

## Result

### Search results

After searching various databases, we were able to retrieve a total of 172 articles. We removed 20 articles due to duplication, and 128 articles were excluded after reviewing the title and abstract. Subsequently, 24 articles were sought for full article retrieval and eligibility review. After a full article review, 6 studies reporting wrong outcomes, 1 study with a full article not retrievable, 10 studies done outside Ethiopia, and 1 study done in a similar hospital as the study included in the final review were excluded from the review. A total of six articles were included in the final review (Fig 1).

### Study characteristics

All of the included primary studies used a cross-sectional study design. Of the studies, 1 was done in Sidama region [21], 1 in Addis Ababa [22], 1 study was done in Oromia region [12], and 3 studies were done in Amhara region [13, 23, 24]. The studies were published between 2015 and 2023 and included a sample size ranging from 200 [12] to 410 [21] individuals. The total number of participants included in the review was 2101. Regarding the method of ascertainment of adherence, all of the included studies used self-reporting by the patient to declare adherence. Four studies used a tool designed by the authors for the study [12, 13, 21, 23] while two studies used the Morisky medication adherence scale-8 [22, 24]. All included studies had good quality based on the NOS score.

### Magnitude of glaucoma medication adherence

We used 6 studies with a total of 2101 participants to estimate the pooled prevalence of magnitude of glaucoma medication adherence. There was a statistically significant high heterogeneity among studies based on $I^2$ ($I^2 = 92.82\%$) and a random effect model with the DerSimonian-Laird method was used to determine the pooled magnitude. The magnitude of adherence among adult glaucoma patients was found to be 49.46% (95% CI; 41.27–57.66) (Fig 2). To explore potential sources of heterogeneity, meta-regression was done using sample size and publication year as covariates and both did not affect the heterogeneity (Table 1). Sub-group analysis was done based on the method used to classify adherence. The pooled magnitude of adherence for studies that used a tool prepared by the author was found to be 51.13% (95% CI; 38.83–63.43) and for studies that used the MMAS-8, it was found to be 46.08% (95% CI; 39.32–52.85) (Fig 3).

### Publication bias

Publication bias was assessed using a funnel plot and Eggers test (p = 0.0596) which revealed there was no publication bias (Fig 4).

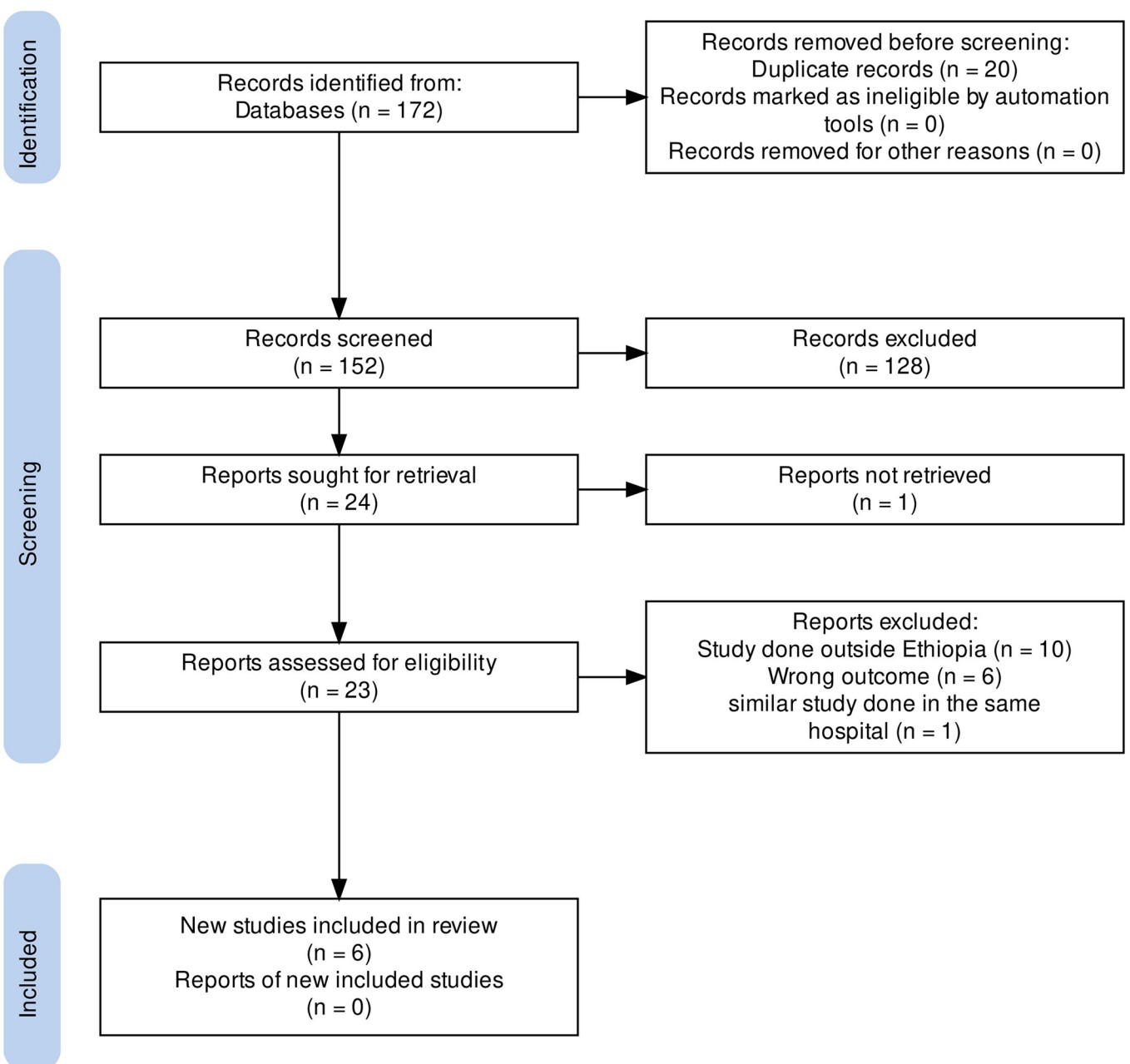

**Fig 1. PRISMA study selection flow for systematic review and meta-analysis of adherence and factors influencing adherence to glaucoma medications among adult glaucoma patients in Ethiopia.**

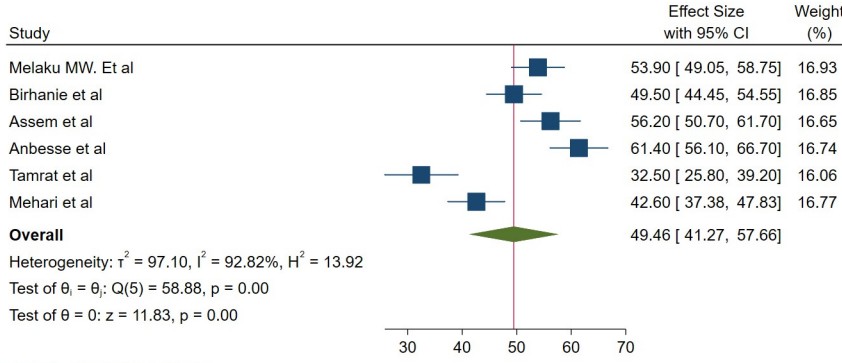

**Fig 2. Forrest plot for the pooled magnitude of adherence to glaucoma medication among adult glaucoma patients in Ethiopia.**

**Table 1. Meta-regression analysis of factors affecting between-study heterogeneity.**

| Source | Coefficient | Std. error | p-value |
|---|---|---|---|
| Sample size | .109635 | 0.076137 | 0.15 |
| Publication year | -0.04151 | 1.763183 | 0.981 |

## Factors influencing adherence to glaucoma medications

We included variables that were significantly associated with adherence in at least two or more of the studies. Summary of factors influencing adherence identified from literatures and not included in the meta-analysis is presented in Table 2.

**Association between place of residence and glaucoma medication adherence.** We used three studies with 1160 participants [13, 21, 23] to assess the association between place of residence and adherence. The odds of adherence to glaucoma medication was found to be 1.89 times higher among urban residents than rural residents (OR = 1.89, 95% CI; 1.29–2.49) (Fig 5).

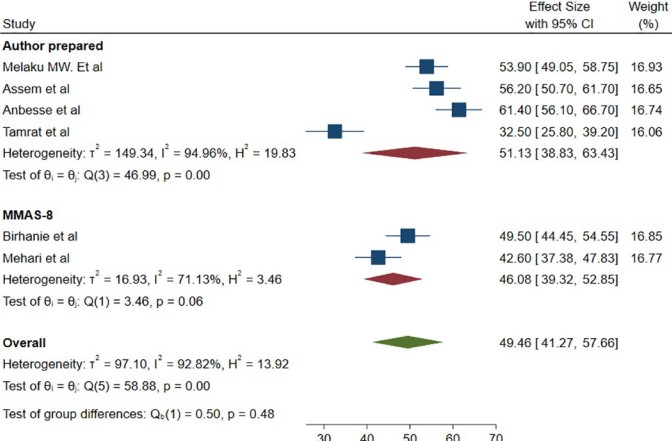

**Fig 3. Forrest plot for the pooled magnitude of adherence to glaucoma medication among adult glaucoma patients based on the method of adherence classification used.**

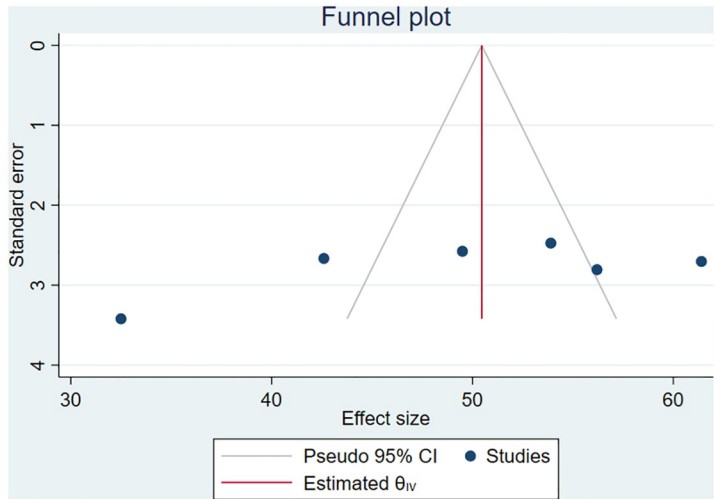

**Fig 4. Funnel plot assessing publication bias of the magnitude of glaucoma medication adherence among adult glaucoma patients in Ethiopia.**

**Association between visual acuity and glaucoma medication adherence.** We used three studies with 1160 participants [13, 21, 23] to assess the association between visual acuity and adherence. The odds of adherence to glaucoma medication was found to be 2.82 times higher among patients with normal visual acuity than those with affected visual acuity (OR = 2.82, 95% CI; 0.85–4.80, P = 0.01) (Fig 6).

**Association between means of payment and glaucoma medication adherence.** We used two studies to examine the association between means of payment for medication and adherence with a total of 719 study participants [13, 22]. It was found that those who pay for the glaucoma medication themselves had 78% lower odds of adherence to glaucoma medication than those who are sponsored (OR = 0.22, 95%CI; 0.09–0.34) (Fig 7). In addition, one study

**Table 2. Study characteristics of included studies on adherence and factors influencing adherence to glaucoma medications among adult glaucoma patients in Ethiopia.**

| Authors/Year | Region | Study design | SS | NOS score | Adherence (95% CI) | Main findings |
|---|---|---|---|---|---|---|
| Melaku M. Et al, 2023 [21] | Sidama | Cross-sectional | 410 | 7 | 53.9% (48.8–58.5) | Factors favoring adherence: Urban residence, higher educational level, monthly follow-up frequency, and normal vision |
| Birhanie et al, 2022 [24] | Amhara | Cross-sectional | 382 | 8 | 49.5% (44.4–54.5) | Factors favoring adherence: good knowledge, favorable attitude, a short distance from homes to hospitals Factors favoring non-adherence: scheduling problems for glaucoma follow-up |
| Assem et al,2020 [23] | Amhara | Cross-sectional | 390 | 7 | 56.2% (51–62) | Factors favoring adherence: Early glaucoma, normal vision, urban residence, having family support, and receiving information from pharmacist |
| Anbesse et al, 2018 [13] | Amhara | Cross-sectional | 360 | 7 | 61.4% (56.1–66.7) | Factors favoring adherence: Male sex, urban residence, normal visual acuity, low visual acuity, and self-sponsor for medications |
| Mehari T. et al, 2016 [22] | Addis Ababa | Cross-sectional | 359 | 7 | 42.6% (37.38–47.83) | Factors favoring adherence: Higher educational level, being self–employed, and taking lesser frequency of drops Factors favoring non-adherence: low monthly family income and self—purchasing of medications |
| Tamrat et al, 2015 [12] | Oromia | Cross-sectional | 200 | 7 | 32.5% (25.8–39.2) | Older age, advanced stage of glaucoma, longer frequency of follow up, and financial problem are associated with non-adherence |

SS; sample size, NOS; Newcastle Ottawa score

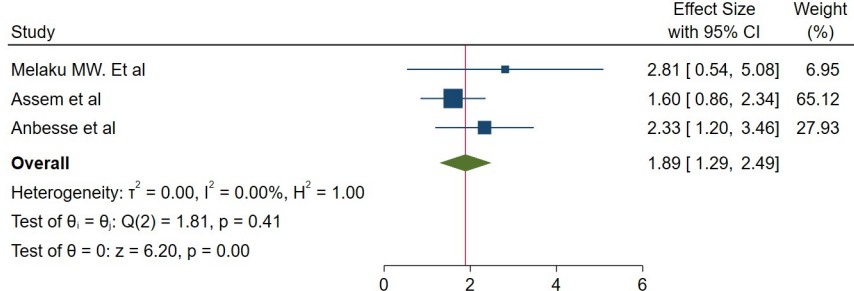

**Fig 5. Association between place of residence and glaucoma medication adherence.**

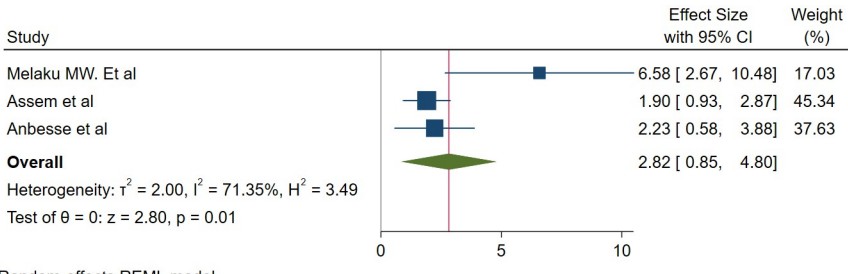

**Fig 6. Association between visual acuity and glaucoma medication adherence.**

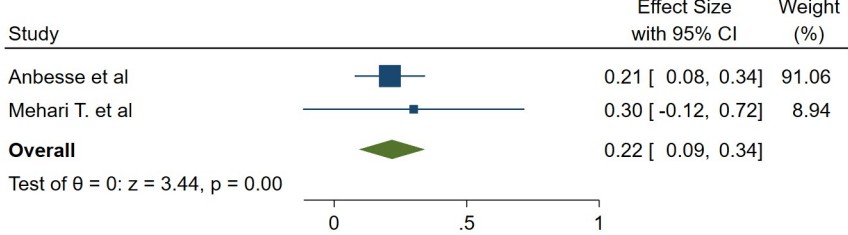

**Fig 7. Association between means of payment and glaucoma medication adherence.**

reported a statistically significant association between financial problems and non-adherence [12]. However, the study did not include the measure of effect, and thus it was not included in the meta-analysis.

## Discussion

In this study, it was found that less than half of glaucoma patients (49.46%) adhered to the prescribed glaucoma medications. This finding is comparable with a study done in Ghana (48.5%) [25], Egypt (46.4%) [26], and higher than reported in Togo (10%) [27] and Iran (34.6%) [16]. However, it is lower than reported in studies conducted in Benin (53%) [28], Israel (71%) [29], Saudi Arabia (72.6%) [30], and Korea (72.6%) [31]. This variation could be due to differences in socioeconomic profiles, differences in the health systems and health insurance coverage of the countries, differences in the definition and method of ascertaining adherence, as well as variations in sample size.

In this study, it was found that those residing in urban areas had higher odds of adherence than those residing in rural areas. A study done in Iran in 2015 also reported a statistically significant association between place of residence and medication adherence [16]. This could be due to differences in awareness, access to health care and health-related information, and differences in access to health services like pharmacies for medication refill [32]. In Ethiopia, glaucoma related services are mainly concentrated in the major cities. Patients from rural areas travel long distances with limited transportation to get services [33]. The cost of transportation and accommodations would also increase the total expenses of patients [34]. This may cause the inaccessibility of glaucoma medications for those coming from rural areas and poor adherence.

Furthermore, those with normal vision had higher odds of adhering to glaucoma medication compared to those with affected vision. This could be related to the dissatisfaction with the lack of change in the disease among those with affected visual acuity [16, 29]. Additionally, those who have impaired visual acuity are highly likely to be dependent on others for their day to day activities including application of the medication, purchasing medication, and going to health facilities for follow-up visits. This could prevent patients from adhering to glaucoma medication. This finding is supported by a study done in Israel that showed a statistically significant association between functional dependence and glaucoma medication adherence (p = 0.005) [29].

Those who paid for the medication by themselves had lower odds of adherence than those who were sponsored for their medication. This finding is in comparative agreement with studies done in Nigeria [35], Ghana [36], and Romania [37] that reported the cost of the medication as a significant barrier to glaucoma medication adherence. Previous studies done in Ethiopia showed that financial concern is one of the major barriers to eye care service utilization [34, 38]. This concern is more pronounced among glaucoma patients due to the long course of treatment and the high cost of glaucoma medications [39]. The financial burden would be even higher for those who are on multiple glaucoma medications [16, 26] and those with comorbidities taking other medications. Furthermore, despite the availability and promotion of community based health insurance in Ethiopia, out-of-pocket expenditures for health care still constitute a higher proportion of health care expenditures [40]. This could make eye care services in general, and glaucoma care in particular, inaccessible and unaffordable.

## Limitations

This study is affected by recall bias and desirability bias inherent to the primary studies that used a self-reporting method of adherence ascertainment. This could potentially lead to an overestimation of medication adherence. Moreover, only 4 out of the 12 regions of Ethiopia are represented in the study, which makes the finding less generalizable to Ethiopia.

## Conclusion

The magnitude of adherence to glaucoma medication is low among adult glaucoma patients in Ethiopia. Tailored health education for those coming from rural areas and those with affected visual acuity regarding glaucoma and drug adherence is recommended. Furthermore, expanding glaucoma related services to rural areas to make the service more accessible and affordable is recommended. Additionally, availing glaucoma medication at a subsidized price and expanding and raising awareness about community based health insurance to make glaucoma medications more accessible and affordable is recommended. We also recommend further studies to be done using objective and validated methods of adherence assessment.

## Supporting information

**S1 Checklist. PRISMA 2020 checklist.**
(DOCX)

**S1 Dataset.**
(XLSX)

## Author Contributions

**Conceptualization:** Kibruyisfaw Weldeab Abore, Estifanos Bekele Fole, Mahlet Tesfaye Abebe.

**Data curation:** Kibruyisfaw Weldeab Abore, Estifanos Bekele Fole, Mahlet Tesfaye Abebe, Natnael Fikadu Tekle, Robel Bayou Tilahun, Fraol Daba Chinkey, Michael Teklehaimanot Abera.

**Formal analysis:** Kibruyisfaw Weldeab Abore, Robel Bayou Tilahun, Michael Teklehaimanot Abera.

**Investigation:** Kibruyisfaw Weldeab Abore, Mahlet Tesfaye Abebe, Natnael Fikadu Tekle, Fraol Daba Chinkey, Michael Teklehaimanot Abera.

**Methodology:** Kibruyisfaw Weldeab Abore, Estifanos Bekele Fole, Mahlet Tesfaye Abebe, Natnael Fikadu Tekle, Robel Bayou Tilahun.

**Resources:** Estifanos Bekele Fole, Mahlet Tesfaye Abebe.

**Software:** Kibruyisfaw Weldeab Abore, Estifanos Bekele Fole, Robel Bayou Tilahun, Fraol Daba Chinkey, Michael Teklehaimanot Abera.

**Validation:** Estifanos Bekele Fole, Natnael Fikadu Tekle, Fraol Daba Chinkey.

**Visualization:** Kibruyisfaw Weldeab Abore, Natnael Fikadu Tekle, Fraol Daba Chinkey.

**Writing – original draft:** Kibruyisfaw Weldeab Abore, Mahlet Tesfaye Abebe, Natnael Fikadu Tekle, Robel Bayou Tilahun, Fraol Daba Chinkey, Michael Teklehaimanot Abera.

**Writing – review & editing:** Kibruyisfaw Weldeab Abore, Estifanos Bekele Fole, Mahlet Tesfaye Abebe, Natnael Fikadu Tekle, Robel Bayou Tilahun, Fraol Daba Chinkey, Michael Teklehaimanot Abera.

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
