## [Decision Letter · Decision Letter 0]

30 Jan 2024

PONE-D-23-32159Glaucoma medication adherence and associated factors among adult glaucoma patients in Ethiopia: A systematic review and meta-analysisPLOS ONE

Dear Dr. Abore,

Thank you for submitting your manuscript to PLOS ONE. After careful consideration, we feel that it has merit but does not fully meet PLOS ONE’s publication criteria as it currently stands. Therefore, we invite you to submit a revised version of the manuscript that addresses the points raised during the review process.

We look forward to receiving your revised manuscript.

Kind regards,

Osamudiamen Cyril Obasuyi, MD, MSc, FWACS, FMCOPh

Academic Editor

PLOS ONE

Journal Requirements:

When submitting your revision, we need you to address these additional requirements. 1. Please ensure that your manuscript meets PLOS ONE's style requirements, including those for file naming. The PLOS ONE style templates can be found at https://journals.plos.org/plosone/s/file?id=wjVg/PLOSOne_formatting_sample_main_body.pdf and https://journals.plos.org/plosone/s/file?id=ba62/PLOSOne_formatting_sample_title_authors_affiliations.pdf 2. Did you know that depositing data in a repository is associated with up to a 25% citation advantage (https://doi.org/10.1371/journal.pone.0230416)? If you’ve not already done so, consider depositing your raw data in a repository to ensure your work is read, appreciated and cited by the largest possible audience. You’ll also earn an Accessible Data icon on your published paper if you deposit your data in any participating repository (https://plos.org/open-science/open-data/#accessible-data). 3. We note that there is identifying data in the Supporting Information file "data set.xlsx
". Due to the inclusion of these potentially identifying data, we have removed this file from your file inventory. Prior to sharing human research participant data, authors should consult with an ethics committee to ensure data are shared in accordance with participant consent and all applicable local laws. Data sharing should never compromise participant privacy. It is therefore not appropriate to publicly share personally identifiable data on human research participants. The following are examples of data that should not be shared: -Name, initials, physical address-Ages more specific than whole numbers-Internet protocol (IP) address-Specific dates (birth dates, death dates, examination dates, etc.)-Contact information such as phone number or email address-Location data-ID numbers that seem specific (long numbers, include initials, titled “Hospital ID”) rather than random (small numbers in numerical order) Data that are not directly identifying may also be inappropriate to share, as in combination they can become identifying. For example, data collected from a small group of participants, vulnerable populations, or private groups should not be shared if they involve indirect identifiers (such as sex, ethnicity, location, etc.) that may risk the identification of study participants. Additional guidance on preparing raw data for publication can be found in our Data Policy (https://journals.plos.org/plosone/s/data-availability#loc-human-research-participant-data-and-other-sensitive-data) and in the following article: http://www.bmj.com/content/340/bmj.c181.long. Please remove or anonymize all personal information (<specific identifying information in file to be removed>), ensure that the data shared are in accordance with participant consent, and re-upload a fully anonymized data set. Please note that spreadsheet columns with personal information must be removed and not hidden as all hidden columns will appear in the published file. 4. Please include your full ethics statement in the ‘Methods’ section of your manuscript file. In your statement, please include the full name of the IRB or ethics committee who approved or waived your study, as well as whether or not you obtained informed written or verbal consent. If consent was waived for your study, please include this information in your statement as well.

Additional Editor Comments:

TITLE: The title is unclear. What associated factors? If the authors wouldn’t mind a slight alteration to the title, I will like to suggest this:

Adherence, and factors influencing adherence to Glaucoma medications among adult glaucoma patients in Ethiopia: A systematic review and meta-analysis.

I think this title better conveys the goals of the authors.

ABSTRACT

1. Glaucoma medications are one important pillar of glaucoma management to control intraocular pressure- The statement is unclear.

2. If left uncontrolled, intraocular pressure causes progressive visual loss and blindness.-This statement is not entirely accurate. While IOP is the only modifiable risk factor, various other factors may contribute to Visual loss in Glaucoma. The statement could be rewritten to reflect the true role of IOP in glaucoma.

3. Results do not properly delineate the factors affecting adherence. Which factors improved or discouraged adherence.

INTRODUCTION

1. Why have the authors decided to carry out this review in Ethiopia? The reason for this review in Ethiopia has not been justified.

2. The authors used "its associated factors" continually throughout the review. This phrase is unclear and does not fully convey the goal of the review. Factors affecting adherence could be positive or negative. The use of the phrase may have colored the way the review may have been carried out.

RESULTS

1. The authors should provide a summary of the various factors affecting adherence-positively or negatively- as studied by the various reports included in this review. This summary provides better understanding of the results.

DISCUSSION and CONCLUSION

The discussion currently reads like a repeat of the results. The authors have done nothing to contextualize the results and the import these results may have in Ethiopia. They have just compared their results with reports from other countries. This has also led to the conclusion that was drawn by the authors.

Reviewers' comments:

Reviewer's Responses to Questions

**Comments to the Author**

1. Is the manuscript technically sound, and do the data support the conclusions?

Reviewer #1: Yes

Reviewer #2: Yes

2. Has the statistical analysis been performed appropriately and rigorously? 

Reviewer #1: Yes

Reviewer #2: Yes

3. Have the authors made all data underlying the findings in their manuscript fully available?

Reviewer #1: Yes

Reviewer #2: Yes

4. Is the manuscript presented in an intelligible fashion and written in standard English?

Reviewer #1: No

Reviewer #2: Yes

5. Review Comments to the Author

Reviewer #1: The presented study is an original systematic review and meta-analysis with included completed PRISMA checklist and flow diagram. This research adheres to appropriate reporting guidelines and meets all main Publication Criteria in PLOS ONE, but there are a few comments:

— In the “Ethics Statement” it should be written “N/A”, because the submission does not require an ethics statement.

— Manuscript should be written in a clear, correct, and unambiguous language. Please correct some punctuation and grammatical errors, like in the Line 43: “……cause of blindness, which ………..”; in the Line 33: “………. for meta-analysis” (without “the”); in the Line 57: Studies with the small “s”; in the Line 63: “vary” instead of “varies”; in the Line 129: Articles with the small “a”; in the Line 220: “…..among those, who are ….“, etc.

— Line 21: “Glaucoma medications are one of the important pillars……”

— Line 22-23: “Intraocular pressure causes progressive visual loss and blindness, if left uncontrolled.”

— In the Section “Introduction” the data on incidence and blindness related to glaucoma should be updated – Line 44-45 - (According to the World Health Organization, the total number of patients with glaucoma is about 105 million people, 11.2 million people are blind from glaucoma.)

— Line 46: Please add the abbreviation “IOP” after intraocular pressure

— In the Section “Results”: How the final review could include 6 articles, if the total number of retrieved articles was 170, 20 articles were removed due to duplication; 144 articles were excluded after reviewing the title and abstract, and additionally 6 studies reporting wrong outcomes were also excluded. Please give more descriptive information, according to the PRISMA flow diagram, which is correct

— Line 141: “Four studies used a tool …….” instead of “Of the studies, 4 of them used …“

— Line 197: “However, it is lower than reported or showed in studies……”

— Line 234: “……methods of adherence assessment…”

Reviewer #2: Study was reasonably well carried out and the limitations highlighted by the authors. Relevant statistical analysis were carried out. Recommendations were made that the authors believe would help in improving compliance with glaucoma medications.

6. PLOS authors have the option to publish the peer review history of their article (what does this mean?). If published, this will include your full peer review and any attached files.

Reviewer #1: No

Reviewer #2: No

---

## [Author Response · Author response to Decision Letter 0]

8 Feb 2024

Dear Editor,

Thank you for the insightful comments we received. We have noted the concerns raised by the reviewers and we have carefully edited the manuscript. We have given our responses below in chronological order. 

Sincerely,

Kibruyisfaw Weldeab (MD, MPH)

Editorial corrections

Author response: Thank you for the comments. We have carefully edited the manuscript as per the suggestion.

2. “We note that there is identifying data in the Supporting Information file "data set.xlsx ". Due to the inclusion of these potentially identifying data, we have removed this file from your file inventory. Prior to sharing human research participant data, authors should consult with an ethics committee to ensure data are shared in accordance with participant consent and all applicable local laws.”

Author response: Thank you for the comments. The identifiers included were the names of primary authors. We have corrected as per the suggestion. 

3. “Please include your full ethics statement in the ‘Methods’ section of your manuscript file. In your statement, please include the full name of the IRB or ethics committee who approved or waived your study, as well as whether or not you obtained informed written or verbal consent. If consent was waived for your study, please include this information in your statement as well”

Author response: Thank you for the comments. We have corrected as per the suggestion.

4. “TITLE: The title is unclear. What associated factors? If the authors wouldn’t mind a slight alteration to the title, I will like to suggest this: Adherence, and factors influencing adherence to Glaucoma medications among adult glaucoma patients in Ethiopia: A systematic review and meta-analysis. I think this title better conveys the goals of the authors.”

Author response: Thank you for the comments. We have corrected as per the suggestion.

5. Glaucoma medications are one important pillar of glaucoma management to control intraocular pressure- The statement is unclear.

Author response: Thank you for the comments. We have edited the abstract to make it clearer.

6. If left uncontrolled, intraocular pressure causes progressive visual loss and blindness.-This statement is not entirely accurate. While IOP is the only modifiable risk factor, various other factors may contribute to Visual loss in Glaucoma. The statement could be rewritten to reflect the true role of IOP in glaucoma.

Author response: we accept the editor’s comment and we have edited the abstract accordingly.

7. Results do not properly delineate the factors affecting adherence. Which factors improved or discouraged adherence?

Author response: Thank you for the comments. We have edited the result to address the editors’ comment.

INTRODUCTION. 

8. Why have the authors decided to carry out this review in Ethiopia? The reason for this review in Ethiopia has not been justified.

Author response: Thank you for the comments. We accept the comment. We have edited the introduction and addressed the concern raised.

9. The authors used "its associated factors" continually throughout the review. This phrase is unclear and does not fully convey the goal of the review. Factors affecting adherence could be positive or negative. The use of the phrase may have colored the way the review may have been carried out.

Author response: Thank you for the comments. We have edited based on the recommendation.

RESULTS. 

10. The authors should provide a summary of the various factors affecting adherence-positively or negatively- as studied by the various reports included in this review. This summary provides better understanding of the results.

Author response: Thank you for the comments. We have corrected as per the suggestion. We have included the factors favoring adherence and the factors favoring non-adherence on table1. 

DISCUSSION and CONCLUSION.

11. “The discussion currently reads like a repeat of the results. The authors have done nothing to contextualize the results and the import these results may have in Ethiopia. They have just compared their results with reports from other countries. This has also led to the conclusion that was drawn by the authors.”

Author response: we sincerely thank the editor for the comment. We have noted the concern and edited the discussion and conclusion section to contextualize to Ethiopia.

Reviewer 1 comments

Thank you for the comments and we have made the following changes based on the comments.

12. In the “Ethics Statement” it should be written “N/A”, because the submission does not require an ethics statement.

Author response: Thank you for the comments. We have corrected as per the suggestion.

13. Manuscript should be written in a clear, correct, and unambiguous language. Please correct some punctuation and grammatical errors, like in the Line 43: “……cause of blindness, which ………..”; in the Line 33: “………. for meta-analysis” (without “the”); in the Line 57: Studies with the small “s”; in the Line 63: “vary” instead of “varies”; in the Line 129: Articles with the small “a”; in the Line 220: “…..among those, who are ….“, etc.

Line 21: “Glaucoma medications are one of the important pillars……” Line 22-23: “Intraocular pressure causes progressive visual loss and blindness, if left uncontrolled.” Line 141: “Four studies used a tool …….” instead of “Of the studies, 4 of them used …“ Line 197: “However, it is lower than reported or showed in studies……”. Line 234: “……methods of adherence assessment…”

Author response: Thank you for the comments. We accept the reviewer’s comments. We have edited each sections of the manuscript using language editing services.

14. In the Section “Introduction” the data on incidence and blindness related to glaucoma should be updated – Line 44-45 - (According to the World Health Organization, the total number of patients with glaucoma is about 105 million people, 11.2 million people are blind from glaucoma.)

Author response: Thank you for the comments. However, we were not able to identify the figure mentioned by reviewer. We have included recent figure from WHO that states 76 million people are affected by glaucoma and projected to be 95.4 million in 2040. Furthermore a factsheet from WHO published in August 2023 indicated 7.7 million people had visual impairment secondary to glaucoma.

15. Line 46: Please add the abbreviation “IOP” after intraocular pressure 

Author response: Thank you for the comments. We have corrected it.

16. In the Section “Results”: How the final review could include 6 articles, if the total number of retrieved articles was 170, 20 articles were removed due to duplication; 144 articles were excluded after reviewing the title and abstract, and additionally 6 studies reporting wrong outcomes were also excluded. Please give more descriptive information, according to the PRISMA flow diagram, which is correct.

Author response: Thank you for the comments. We have corrected the error we committed during write up and rewritten it based on the PRISMA flow diagram

Reviewer 2 comments

Thank you for the comments.

---

## [Editor Report · Decision Letter 1]

12 Feb 2024

Adherence and factors influencing adherence to glaucoma medications among adult glaucoma patients in Ethiopia: A systematic review and meta-analysis

PONE-D-23-32159R1

Dear Dr. Abore,

We’re pleased to inform you that your manuscript has been judged scientifically suitable for publication and will be formally accepted for publication once it meets all outstanding technical requirements.

Kind regards,

Osamudiamen Cyril Obasuyi, MD, MSc, FWACS, FMCOPh

Academic Editor

PLOS ONE
---

## [Editor Report · Acceptance letter]

1 Mar 2024

PONE-D-23-32159R1 

PLOS ONE

Dear Dr. Abore, 

I'm pleased to inform you that your manuscript has been deemed suitable for publication in PLOS ONE. Congratulations! Your manuscript is now being handed over to our production team.

Kind regards, 

on behalf of

Dr. Osamudiamen Cyril Obasuyi 

Academic Editor

PLOS ONE